# Gluten Conformation at Different Temperatures and Additive Treatments

**DOI:** 10.3390/foods11030430

**Published:** 2022-02-01

**Authors:** Pavalee Chompoorat, Ayuba Fasasi, Barry K. Lavine, Patricia Rayas-Duarte

**Affiliations:** 1Faculty of Engineering and Agro-Industry, Maejo University, Chiang Mai 50290, Thailand; pavalee@mju.ac.th; 2Department of Chemistry, Oklahoma State University, Stillwater, OK 74078, USA; ayubafasasi@gmail.com (A.F.); barry.lavine@okstate.edu (B.K.L.); 3Department of Biochemistry and Molecular Biology, Robert M Kerr Food & Agricultural Products Center, Oklahoma State University, Stillwater, OK 74078, USA

**Keywords:** gluten, diacetyl tartaric acid ester of monoglycerides, dithiothreitol, ascorbic acid, temperature, FTIR spectroscopy

## Abstract

The effect of temperature (25, 45, and 65 °C) on the gluten secondary structure was investigated by using Fourier transform infrared (FTIR) spectroscopy and modulation of disulfide and hydrogen bonds contributions (100 ppm ascorbic acid (AA), 0.6% diacetyl tartaric acid ester of monoglycerides (DATEM), and 0.25 mM dithiothreitol (DTT)). The results showed that additives heated at 65 °C altered most of the gluten matrix formation by changing structural secondary structures compared to the secondary structures of native gluten (control). The content of random coils, α-helices, and β-sheet of gluten increased, while the extent of β-turns and antiparallel β-sheets decreased, which led to the transformation to a more stable secondary conformation. In addition, the rheological properties (%creep strain) revealed that gluten deformation increased during the heating process with all of the additives. The chemometric method could quantitate an overall alteration of gluten polymerization and gluten matrix formation during heating with additive treatments.

## 1. Introduction

As one of the largest biopolymers in nature, the gluten protein is a complex system with multi-components and specific aggregation properties. Gluten is a key component in wheat-based products for their desirable requirements, such as viscoelastic properties, water absorption capacity, and gas holding ability, through its three-dimensional matrix. The intermolecular bonding of glutenin was applied to group the gluten structure into repetitive domains (spiral structure by β-turn) and non-repetitive domains (α-helices structure and other cysteine residues). Thus, hydrogen bonding could impact the secondary structure of gluten. The cysteines in the C-terminal domain of α- and β- gliadins and ω-gliadins provide three and four homologous intrachain disulfide bonds of gliadin, respectively [1,2,3,4,5,6,7,8]. Monomeric gliadin protein with a low molecular weight (28–50 kDa) and S-poor ω-type does not provide elastic properties.

The structural changes of gluten upon heating have been previously studied due to its importance in wheat-based products. Heat-induced changes and various additive treatments impact gluten protein conformation during the breadmaking process in the heat range of 30 to 86 °C. Protein interactions are modified by disrupting hydrogen bonds, disulfide bonds, and hydrophobic interactions [9,10]. At around 45 to 50 °C, a decrease in β-sheet and α-helix structures was observed in both glutenins and gliadins, as well as an increase in irreversible crosslinks in the gluten [11,12]. In addition, heating increases in the β-turns of gliadin resulted in changes of conformation at 20 °C to 80 °C [10].

DATEM is an effective surfactant that can increase proving and loaf volume by enhancing the kinetic stability of gluten and dough systems, reducing gas bubbles size in bread, promoting protein−starch−lipid interactions, which lead to an increase in resistance to deformation and breadmaking functionality [13,14,15,16,17]. Ascorbic acid (AA) is a dough improver and increases dough strength by oxidizing disulfide linkages, which are related to the improved gas retention ability during fermentation and baking [7]. Changes in the secondary structure of gluten and dough can also be studied using an antagonistic approach with dithiothreitol (DTT). DTT is a reducing agent and disrupts disulfide bonds, and therefore can be used as a tool to study the qualitative and quantitative changes in the secondary structure of gluten and dough. A distinct pattern of viscoelastic changes is observed as gluten is oxidized, reduced, or disrupted by hydrogen bonds compared to the surfactant/emulsifier, with the latter increasing the elasticity and viscoelasticity, as well as decreasing viscosity, using DATEM, AA, urea, and DTT [18].

FTIR spectroscopy has been used to study conformational changes from dough processing, such as kneading and stretching, which showed an increase in β-sheet structure and a decrease in α-helices and β-turn [19,20]. Recently, this technique was applied to examine the structure of gluten after being heated with a microwave. The secondary structure of gluten began to be altered after exposure to 1000 W of microwave power by promoting α-helices and β-turns [21], and contributed to the formation of disulfide or isopeptide bonds [22]. Moreover, FTIR also revealed that ultrasonic treatment changed the secondary structure of gluten through the conversion of β-turns to β-sheet [23]. Thus, FTIR is a sensitive technique and can yield valuable insights regarding changes in the gluten structure. In this work, we investigated the effect of temperature and additives on the secondary structure of gluten using FTIR. The objective of this study was to estimate the degree of structural alterations directly from heat and chemical modifications, and to propose possible mechanisms responsible for changes in the gluten secondary structures due to these treatments.

## 2. Materials and Methods

### 2.1. Gluten Preparation

The commercial bread flour of hard red winter (HRW) wheat (*Triticum aestivum* L.) was obtained from Shawnee Milling Co. (Shawnee, OK, USA), and was used as a base flour. The protein, moisture, and ash content were analyzed using a Foss System model 6500 (Foss NIR System Inc., Laurel, MD, USA). The control was native gluten extracted from the dough with no heat (25 °C) or with additive treatment. Three flour treatments and levels were selected (0.6% DATEM, 100 ppm AA, and 25 mM DTT) from previous studies [18]. The selected concentration levels were based on their optimum breadmaking performance and viscoelastic properties from a study comparing four levels of each additive [18]. DATEM (surfactant) and AA (oxidizing/reducing agent) are additives used in baking for improving dough handling, bread volume, and texture, while DTT reduces the disulfide bonds of the gluten proteins. Solutions of DATEM, AA, and DTT were prepared in order to deliver the desired concentration in 0.5 mL aliquot added to the flour. For DATEM, 5 mL DATEM solution (1.2 g DATEM in 100 mL of 2% NaCl solution) was dissolved by sonication and heat (65 °C). Aliquots of 0.5 mL of DATEM solution were added directly to 10 g of flour prior to mixing in a Glutomatic (model 2202, Perten Instruments, Huddinge, Sweden) so as to extract the wet gluten. The dough was mixed for 20 s before washing soluble compounds with 2% NaCl solution through an 88 µm polyester membrane for 5 min. The remaining wet gluten in the chamber was used as described in the following step.

### 2.2. Creep Test with Temperature

The wet gluten obtained from the Glutomatic was immediately rolled into a ball shape and allowed to relax in a 2.5 kg top plate for one hour at room temperature. The gluten was cut into a 25 mm diameter disk using a round metal die, and was transferred to the sample plate of the rheometer. When the gluten was between the upper and lower plate of the rheometer, additional trimming was applied as needed for the size of the gluten in order to match the upper plate diameter. To prevent moisture loss, a small amount of mineral oil was applied to the edge of gluten. An AR1000 rheometer (TA Instruments, New Castle, DE, USA) was used and the test consisted of 100 Pa shear stress (previously tested within linear viscoelastic region) for 100 s for the creep phase. A Peltier temperature controller of the rheometer was used for applying 25, 45, and 65 °C throughout the duration of the test. The percentage of creep strain at 100 s was recorded to demonstrate the deformation of gluten after being subjected to shear stress and temperature. The gluten was freeze-dried immediately after the creep test to preserve its structural changes and for use in the FTIR spectroscopy analysis. The freeze-dried gluten samples were ground using a coffee grinder and were kept in a dry place inside a tightly closed container before testing with FTIR.

### 2.3. Fourier Transform Infrared (FTIR) Spectroscopy

FTIR spectroscopy was used to study the changes in the secondary structures of the gluten. Freeze-dried gluten (100 mg) was hydrated to 47% moisture content using deionized water and was mixed with a spatula to form the hydrated gluten. This moisture content was selected for the highest change in gluten secondary structure, as reported by Georget and Belton [24]. A Nicolet iS50 FTIR spectrometer (Thermo Scientific, Madison, WI, USA) was used to calculate the Fourier self-deconvoluted spectra. The setting was adjusted to acquire spectra in the 1725 cm^−1^ to 1580 cm^−1^ spectral region using a 1.3 enhancement factor and 55 bandwidths. Baseline correction was performed using OMNIC software (Thermo Electron Corp., Madison WI, Scientific, Waltham, MA, USA). Peak assignment for the secondary structure analysis was done following published work on gluten samples [24]. The second derivative was calculated for the spectral region of 1725 to 1580 cm^−1^ using a Savitzky−Golay filtering routine. A 15-data point window and a polynomial order of three were used, as this condition yielded a higher signal to noise derivative spectra. The second derivative of the amide band in the spectral region of 1725 to 1580 cm^−1^ revealed features or valleys that could be associated with the secondary structure of the amide I band. After identifying the potential secondary structures, a curve fitting procedure was applied to the amide I peak (1725 to 1580 cm^−1^) using OMNIC software in order to obtain quantitative estimations of each secondary structural component of the gluten proteins investigated. The area under the curve of each secondary structural component was calculated for each treatment. The band ranges from the amide I band for the protein conformation were as follows: strongly hydrogen-bonded β-sheets, extended hydrated chains, and intermolecular β-sheets (1612–1614 cm^−1^); antiparallel β-sheets (1629–1632 cm^−1^); random coils and α-helices (1650–1660 cm^−1^); and β-turns (1666–1670 cm^−1^).

### 2.4. Statistical Analysis

Analysis of variance (ANOVA) was applied to the data with a significance level of *p* < 0.05 using SPSS 19.0 (SPSS, Inc., Chicago, IL, USA). The results were analyzed using Tukey’s honest significance difference post hoc test at 0.05 probability level for a mean (*n* = 3) comparison of each treatment. Pearson correlation analysis (*p* = 0.05) was carried out for testing the relationship among the secondary structure contents (intermolecular β sheets, antiparallel β-sheets, random coils, β-turns, and β-sheet) and viscoelastic properties (%Creep strain) using XLSTAT v2020 (Long Island City, NY, USA). Principal component analysis (PCA) was conducted on the correlation matrix of the secondary structure contents and viscoelastic properties to reveal the variation and capture the patterns as a function of temperature (25, 45, and 65 °C) and additive treatments (DATEM, AA, and DTT). PCA was performed and the results were depicted in biplot graphs with XLSTAT v2020 (Long Island City, NY, USA).

## 3. Results and Discussion

The amide I bands (1725 to 1580 cm^−1^) of gluten control was shown in Figure 1. The second derivative of amide band provided features that can be related to secondary structure of amide I. Generally, amide I vibrational band absorbs in the range of 1600–1700 cm^−1^, occurred mainly from C=O stretching vibration. This amide I vibration is normally applied for the study of protein backbone structure which is therefore related to spectrum and secondary structure measurement. The spectra were similar to previously reported studies [24]. The second derivative of FTIR spectra of gluten with and without treatments was obtained to enhance structural peaks that were unidentifiable in normal FTIR scan [25]. The band ranges from amide I band for protein conformation were: strongly hydrogen-bonded β-sheets, extended hydrated chains, intermolecular β-sheets (1612–1614 cm^−1^); antiparallel β-sheets (1629–1632 cm^−1^); random coils and α-helices (1650–1660 cm^−1^); and β-turns (1666–1670 cm^−1^) [24].

The amide I bands of gluten at different temperatures and with the addition of different compounds to modulate their disulfide and hydrogen bonds are shown in Figure 2, Figure 3 and Figure 4. The qualitative similarity in secondary derivative FTIR pattern between control and treated samples suggested that in most gluten systems, no new protein conformations were formed (Figure 1, Figure 2, Figure 3 and Figure 4). We hypothesized that the overall secondary structures of gluten subjected to heat and additives were still as functional as the native gluten. However, several changes in gluten secondary structures could be inferred upon examination of the area under the curve (%), especially for random coils and α-helix structures (1649–1650 cm^−1^), and extended chain, hydrogen bonding of β-sheet, and intermolecular β-sheet (1613–1614 cm^−1^), as explained in the following. The percentages of area of each secondary structure component based on the curve fitting procedure and %creep strain of gluten treated with AA, DATEM, and DTT and temperature are shown in Table 1. The secondary structure contents of the control gluten were 26.4% intermolecular β-sheet, 24.3% antiparallel β-sheet, 23.5% random coils and α-helices, 5.6% β-turn, and 8.7% β-sheet (Table 1). In Appendix A, the area (%) of the secondary structures was blocked by temperature for a quick visual reference of the control and temperature/treatments.

### 3.1. Effect of Ascorbic Acid (AA) and Temperature on Gluten Secondary Structural and Viscoelastic Properties

As a bread improver, ascorbic acid (AA) can help increase the loaf volume and improve the crumb structure. The oxidized form of AA, dehydro-L-ascorbic acid (DA), was involved in disulfide bond formation in the presence of oxygen [26]. In this part of the study, AA was added to the gluten system to show how it affected the gluten secondary structure at different temperatures. Figure 2 depicts the amide I band of the hydrated gluten with AA through heating from 25 to 65 °C. All peaks revealed the same amide I band for all secondary structural components as in the native control gluten. The effect of AA on the gluten secondary structure was temperature dependent (Table 1). When gluten with 100 ppm AA was heated, the content of random coils, α-helices, and β-sheet increased, while the intermolecular β-sheet, antiparallel β-sheet, and β-turns decreased at a higher temperature. Thus, gluten conformation was altered to be more folded and compact, as α-helices are more compact than other secondary structures [27]. The percentage reduction of β-turns revealed an enhancement of the disulfide bond and impaired hydrogen bonds [28]. In addition, there was no significant difference in the content of the antiparallel β-sheet and β-turns between the control and gluten with AA at 25 °C. This suggests that AA at 65 °C enhanced the deformation of gluten with a lower content of intermolecular β-sheet, antiparallel β-sheet, and β-turns, but a higher content of random coils, α-helices, and β-sheet compared to gluten with AA at 25 °C. Others have postulated that the oxidant effect could result in an increase in the extended structure, which is consistent with our observations [29]. Additionally, the observed structural changes may be due to a decrease in hydrogen bonding, as observed by the increase in the deformability of gluten treated with AA at higher temperatures using a creep test [30], and evidenced by the 37% increase of %creep strain when heated from 25 to 65 °C (Table 1).

### 3.2. Effect of DATEM and Temperature on Gluten Secondary Structural and Viscoelastic Properties

An anionic surfactant such as DATEM can be incorporated into the gluten protein by using its lipophilic moiety to interact with hydrophobic patches of gluten [27]. The surface charge neutralization due to the presence of DATEM promotes protein conglomeration, which resulted in increasing the specific volume [27]. In our study, the treatment of 0.6% DATEM suggested that it affected all of the secondary structural components of gluten (Table 1). DATEM also improved the gluten deformation by reducing the %creep strain. Thus, the gluten with DATEM was stronger compared to the control. DATEM improved the gluten viscosity element, while it decreased the elasticity and viscoelasticity, as explained by Burger models in a previous study [18]. It encourages protein–starch–lipid interactions, which improve the resistance to flow properties. The addition of DATEM with heating from 25 °C to 65 °C caused an increase in the intermolecular β-sheet (25.5% to 26.3%) and β-sheet contents (6.5% to 8.3%). However, there was no significant difference in random coils and α-helices after heating from 25 °C to 65 °C. This surprising observation for the 45 °C DATEM sample was also confirmed at 1614 cm^−1^ (intermolecular β-sheet), in which its peak (%area) was closely related to the extended chain, hydrogen bonding of β-sheet, and intermolecular β-sheet of the native gluten. Although DATEM was shown to interact with gluten protein initially at 25 °C, we hypothesized that DATEM maybe dissociated from the gluten protein at 45 °C, which caused it to have a similar content of intermolecular β-sheet, random coils, and α-helices when compared with the native gluten at 25 °C. Upon further heating to 65 °C, DATEM may be able to bind to new regions of the gluten surface that were exposed due to the heat. β-turns and antiparallel β-sheets formed a less stable protein conformation [31]. Thus, the secondary structure of gluten with DATEM at 65 °C was more compact by increasing the intermolecular β-sheet and β-sheet contents, and decreasing the antiparallel β-sheets and β-turns compared to 25 °C.

### 3.3. Effect of DTT and Temperature on Gluten Secondary Structural and Viscoelastic Properties

DTT, a reducing agent, can be used to demonstrate the effect of the disulfide bond dissociation on the structure of gluten [7]. Disulfide linkages are considered important contributors to the viscoelastic properties of gluten, in addition to many other factors such as hydrogen bonds, hydrophobic interactions, and the characteristics of the gluten crosslinks [32]. To understand how the loss of the disulfide bond affects the secondary structure, we treated the gluten with 25 mM DTT and studied the FTIR spectral pattern. Gluten with DTT additive at 25 °C had a β-turn structure area comparable to the control native gluten (Table 1). Our results suggest that the loss of disulfide linkages did not significantly alter β-turns at 25 °C. This was consistent with the fact that no covalent bonds were involved in secondary structure conformation. Upon further heating, β-turn structures were significantly different at 45 °C and continued their transformation at 65 °C. Our results suggest that the random coils, α-helices, and β-sheets of the gluten system with DTT underwent a transition as a function of temperature through various conformations. The change in the 1650 cm^−1^ (random coils and α-helices) peak of the DATEM sample may be due to protein rearrangement, which increased the α-helical content, as observed by others [33]. It was suspected that the observed changes at a high temperature were due to the partial loss of the α-helical content [10]. The study of the N-terminal domain of HMW 1Dx5 with heat and salts also found that heat altered the transition of α-helix to β-sheet conformation [34]. Our results reveal that the rheological properties of gluten with DTT at 25 °C significantly reduced the %creep strain and deformation. After heating up to 65 °C, gluten deformation increased with DTT, showing the change of gluten network structure with an increase in α-helices. The α-helix was found to be more hydrophobic and rigid due to the lower accessibility of water compared to for the β-sheet. Thus, the higher content of α-helix had less dough stability and was negatively correlated with dough viscoelasticity [35].

### 3.4. Effect of Temperature and Additive Treatments on Variation of Secondary Structural Motifs and Viscoelastic Properties

The impact of different temperatures (25, 45, and 65 °C) and additive treatments (AA, DATEM, and DTT) on the secondary structural motifs and viscoelastic properties was depicted in a biplot graph by principal component analysis (PCA). The test was used to present sample grouping and to quantitatively evaluate the relationship based on the chemometrics of the parameters shown in Figure 5. The total explained variance was 70.4%. The first principal component (PC1) accounted for 45.6% of the total explained variance, while the second principal component (PC2) accounted for 24.8%. The main contributors for PC1 were %creep strain, β-turns, and antiparallel β-sheets, with 82.6, 72.4, and 70.5%, respectively. Intermolecular β-sheets and β-sheet were highly significant to PC2. The biplot graph provides the relationship between variables, samples, and among variables and samples. The creep strain of the gluten was negatively correlated with antiparallel β-sheets (r = 0.77, *p* = 0.05), as they were in the opposite direction in Figure 5 and Table 2. The variables in a 90° direction means that they are independent from each other. Thus, intermolecular β-sheets are independent from β-sheets, along with random coils. However, the study of dough secondary structure revealed that the increase in β-sheet conformation could improve the intermolecular interactions by hydrogen bonds, which have been accepted to be positively correlated to dough viscoelastic properties [36]. Based on Pearson correlation, the β-turns were negatively correlated with %creep strain and β-sheets (r = −0.67 and −0.66, respectively). Overall, the gluten with additives and heating had different gluten secondary structural components and %creep strain, which led to a different extent of gluten polymerization. Gluten samples with 0.6% DATEM were in quadrant three, closely related to antiparallel β-sheets and random coils, while the samples with 25 mM DTT were in the middle between samples with 0.6% DATEM and 100 ppm AA heated at 65 °C. After heating the gluten with DTT, it was highly correlated with β-turns and antiparallel β-sheets at 25 °C, and then had a high correlation with β-sheet at 45 °C. After increasing the temperature to 65 °C, the gluten with DTT had moved to be close to the %creep strain, which means it had more deformation after heating. AA gluten samples were related to intermolecular β-sheets at 25 °C, and had a higher correlation with %creep strain after heating at 65 °C compared to other treatments.

## 4. Conclusions

The effect of additives and temperature on the secondary structures of gluten was investigated in this study. Overall, additives and temperature yielded differences in %area of gluten secondary structural motifs and %creep strain. Upon heating up to 65 °C, the secondary structures of gluten were revealed to be more compact with all additives, as observed by the increasing of content of random coils, α-helices, and β-sheets. In addition, β-turns and antiparallel β-sheets were reduced, which also contributed to a more stable conformation. The chemometric technique revealed that antiparallel β-sheets were negatively correlated to %creep strain. In addition, random coils, α-helices, and β-sheets were not correlated to %creep strain. A curve fitting of the amide I band for the gluten secondary structure could gain promising information for the quantitating degree of structural alterations from heat and chemical modifications in the gluten structure. The presence of water vapor in the spectra might be a confounding variable. Thus, this matter should be focused on in future work. Future directions include complementing studies such as micro-rheology and thermal analysis to ascertain when structural changes can be detected in such material properties.

## Figures and Tables

**Figure 1 foods-11-00430-f001:**
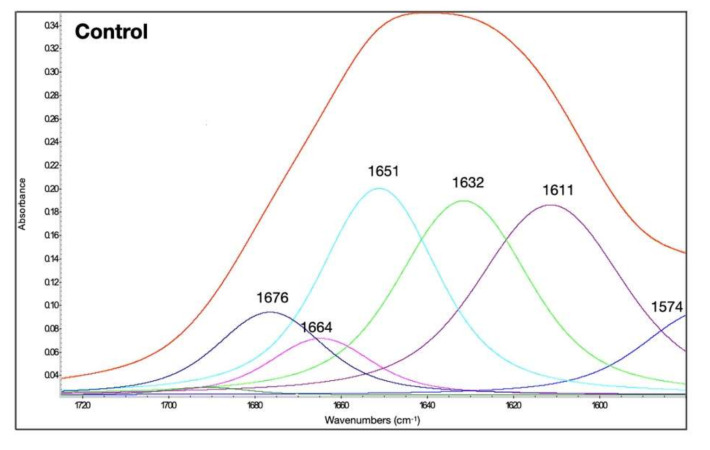
Amide I band of the native gluten control without temperature (25 °C) and the additive treatments. The secondary structure determination by the curve-fitting procedure is indicated.

**Figure 2 foods-11-00430-f002:**
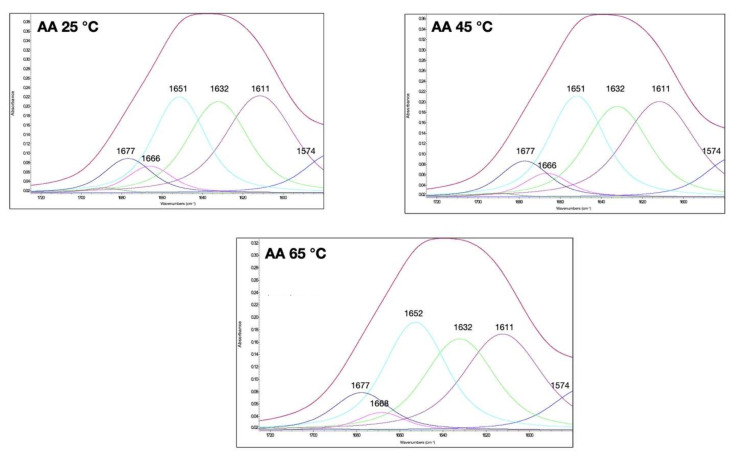
Amide I band of gluten at 25, 45, and 65 °C and with ascorbic acid (AA; 100 ppm) treatment. The secondary structure determination by the curve-fitting procedure is indicated.

**Figure 3 foods-11-00430-f003:**
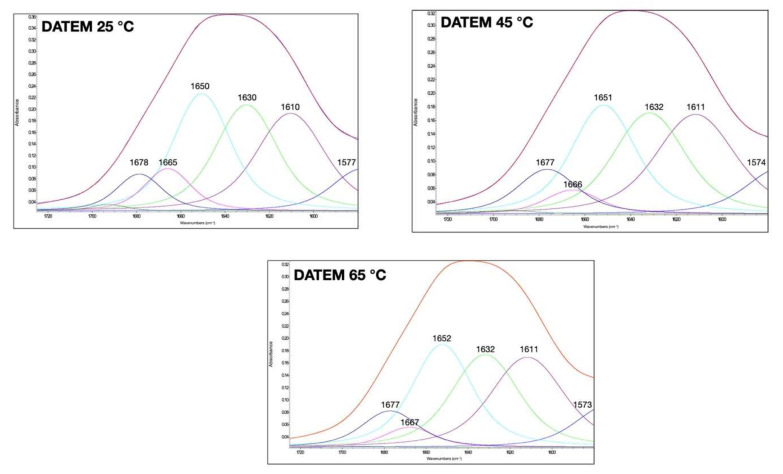
Amide I band of gluten at 25, 45, and 65 °C and with diacetyl tartaric acid ester of monoglycerides (DATEM; 0.6%) treatment. The secondary structure determination by the curve-fitting procedure is indicated.

**Figure 4 foods-11-00430-f004:**
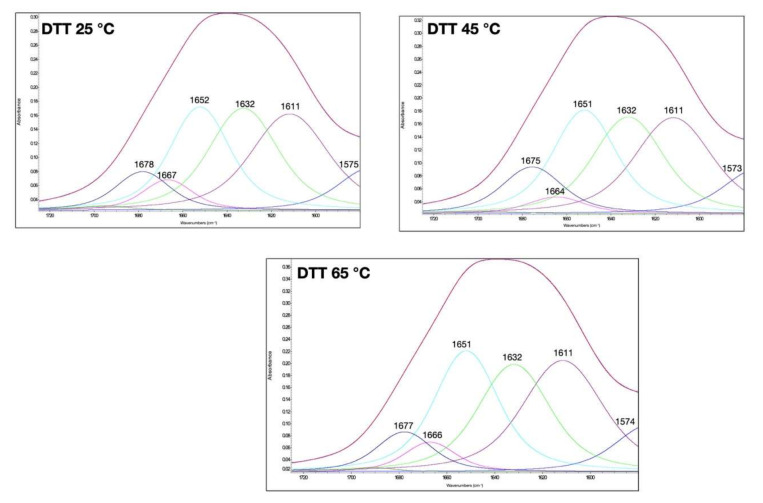
Amide I band of gluten at 25, 45, and 65 °C and with dithiothreitol (DTT) treatment. The secondary structure determination by the curve-fitting procedure is indicated.

**Figure 5 foods-11-00430-f005:**
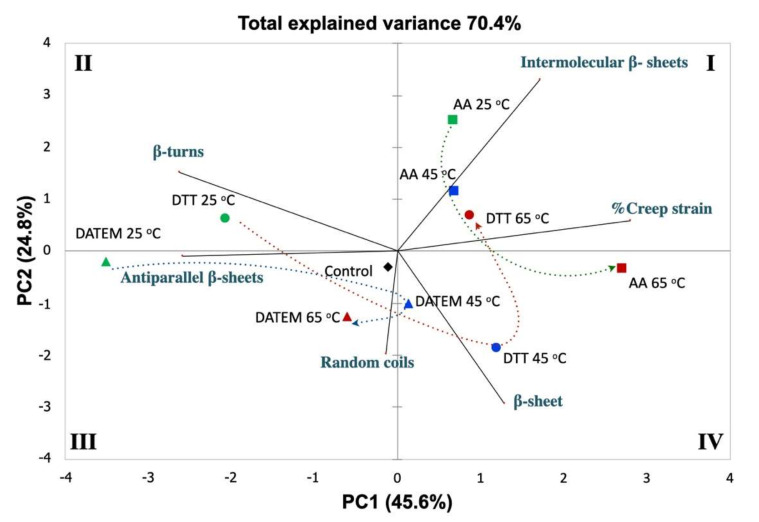
Principal component analysis of gluten with six indicators of secondary structural motifs and the viscoelastic property (creep strain) showing the effect of temperature (25, 45, and 65 °C) and additive (AA, DATEM, and DTT) treatments. Treatments: 25 °C, green; 45 °C, blue; 65 °C, red; AA, square; DATEM, triangle; and DTT, circle. I-IV are graph quadrants in the Cartesian plane.

**Table 1 foods-11-00430-t001:** Gluten secondary structure (FTIR spectroscopy) expressed in area (%) and creep strain (%) as a function of temperature, ascorbic acid (AA), diacetyl tartaric acid ester of monoglycerides (DATEM), and dithiothreitol (DTT) ^a^.

Treatment	Area (%)					Creep Strain (%)
	Intermolecular β-Sheets/ Extended Hydrated Chains	Antiparallel β-Sheets, More Weakly Hydrogen-Bonded β-Sheets	Random Coils and α-Helices	β-Turns	β-Sheets	
Control	26.4 ^f^	24.3 ^d^	23.5 ^e^	5.6 ^b^	8.7 ^c^	13.4 ^d^
AA 25 °C	29.5 ^a^	24.3 ^de^	22.3 ^g^	5.3 ^b^	7.3 ^h^	13.9 ^c^
AA 45 °C	28.9 ^b^	24.2 ^f^	24.0 ^cd^	4.6 ^c^	7.6 ^g^	12.2 ^f^
AA 65 °C	27.9 ^c^	23.8 ^g^	25.7 ^a^	3.1 ^e^	8.2 ^e^	19.1 ^a^
DATEM 25 °C	25.5 ^h^	25.0 ^b^	25.7 ^ab^	7.5 ^a^	6.5 ^j^	7.1 ^i^
DATEM 45 °C	26.5 ^f^	24.3 ^e^	23.8 ^de^	4.4 ^cd^	9.2 ^b^	11.0 ^g^
DATEM 65 °C	26.3 ^g^	24.9 ^c^	25.2 ^b^	4.1 ^d^	8.3 ^d^	11.0 ^g^
DTT 25 °C	27.1 ^e^	25.6 ^a^	22.9 ^f^	5.6 ^b^	7.8 ^f^	9.5 ^h^
DTT 45 °C	26.4 ^f^	24.2 ^f^	23.8 ^de^	3.4 ^e^	10.4 ^a^	12.4 ^e^
DTT 65 °C	27.5 ^d^	23.8 ^g^	24.5 ^c^	4.8 ^c^	7.1 ^i^	15.0 ^b^

^a^ Mean values (*n* = 3) within the same column followed by the same superscript letter are not significantly different (Tukey’s test *p* < 0.05). Treatments: 100 ppm AA, 0.6% DATEM, and 25 mM DTT.

**Table 2 foods-11-00430-t002:** Pearson correlation coefficients between different secondary structures of hydrated gluten and the extracted gluten viscoelastic property (creep strain).

	Intermolecular β-Sheets/Extended Hydrated Chains	Antiparallel β-Sheets, More Weakly Hydrogen-Bonded β-Sheets	Random Coils and α-Helices	β-Turns	β-Sheet	Creep Strain (%)
Intermolecular β- sheets	1.000					
Antiparallel β-sheets	−0.400	1.000				
Random coils	−0.438	−0.158	1.000			
β-turns	−0.262	0.540	−0.063	1.000		
β-sheet	−0.267	−0.200	−0.178	−0.657 *	1.000	
Creep strain	0.547	−0.770 *	0.060	−0.670 *	0.149	1.000

* Significant correlations at *p* < 0.05.

## Data Availability

All data presented is available at request from corresponding author.

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
