# Peer review of "Gluten Conformation at Different Temperatures and Additive Treatments"

_foods, 2022, doi:10.3390/foods11030430_

Round 1

Reviewer 1 Report

The reviewed work concerns research on changes in the structure of gluten during changes in secondary structure changes. The authors use a simple FTIR technique for this purpose. The results showed that additives heated at 65 oC altered most of the gluten matrix formation by changing structural secondary structures compared to secondary structures of control gluten. The obtained results were subjected to chemometric analysis which confirms the FTIR observations. Interesting work, which can help block the allergic properties of gluten (for patients with celiac disease).

I have one comment. Perhaps the authors would be tempted to present the FTIR spectrum changes e.g. for one temperature in one figure (control and addition: AA, DTT, DATEM). It will be more readable.

Reviewer 2 Report

The authors in this study investigated the viscoelastic properties of gluten in bread with respect to temperature changes (25, 45, and 60°C) and the use of additives (DATEM, AA, and DTT) by FTIR. The manuscript looks quite interesting but lacks some important information. I have some comments.

Comment#1: The authors have used commercial bread flour of ‘hard red winter wheat’ that is in my opinion  Triticum aestivum wheat. The authors should mention the correct type of wheat.  

Comment#2: The authors have not mentioned what would be the consequences of heat on wheat?

  • Will viscoelastic properties, water absorption capacity, and gas holding ability compromise? What would be the visible effect?
  • Is such bread will not be safe for the general public or it will be unsafe/more suitable for some specific people e.g. patients with gluten-related disorders?

Comment#3: Overall the FTIR procedure looks fine. But I found several spelling/writing mistakes. Some are as follow:

  • Page 3 - FTIR heading -FTIR spectroscopy FTIR was used to study change it to FTIR Spectroscopy.
  • Page 3- FTIR heading - Amide is written amid at few places- correct it.
  • Page 3- heading FTIR last line - β-turn (1670-1666 cm−1) change it to β-turn (1666-1670 cm−1)
  • Page 4- Result and discussion-first paragraph - FITR change it to FTIR.

Reviewer 3 Report

Dear Authors,

This paper is scientifically sound, well presented, fluent, and robust. Methodologies are at the state of the science, as well as data analyses and statistical procedures. Additionally the recipient research was under the aegis of USDA National Institute of Food and Agriculture.

There’s just one big problem: I cannot see any tables or figures.

So I suggest the authors to resubmit the work entirely.

Some minors:

Introduction is a bit too long and should be cut.

38: 28,000-50,000 daltons…please use kDA

43 and on: 260°C..please revise in 260 °C

92: 100 ml…please correct in 100 mL

Conclusion is poor and must be improved, including weakness and strengths of this work, as well as to introduce some future investigations, perspectives, and applications.

Round 2

Reviewer 3 Report

Dear Authors, thanks for addressing all my suggestions.